# Effects of Crucian Carp (*Carassius auratus*) on Water Quality in Aquatic Ecosystems: An Experimental Mesocosm Study

**Yehui Huang** [1,†]**, Xueying Mei** [2,†]**, Lars G. Rudstam** [3] **, William D. Taylor** [4]**, Jotaro Urabe** [5]**, Erik Jeppesen** [6,7,8] **, Zhengwen Liu** [1,7,9,*] **and Xiufeng Zhang** [1,*]

1 Department of Ecology and Institute of Hydrobiology, Jinan University, Guangzhou 510632, China; eduhyh@163.com
2 College of Resources and Environment, Anhui Agricultural University, Hefei 230036, China; qxxmxy@163.com
3 Department of Natural Resources, Cornell Biological Field Station, Cornell University, New York, NY 13030, USA; rudstam@cornell.edu
4 Department of Biology, University of Waterloo, Waterloo, ON N2L 3G1, Canada; wdtaylor@uwaterloo.ca
5 Graduate School of Life Sciences, Tohoku University, Sendai 980-8577, Japan; urabe@tohoku.ac.jp
6 Department of Bioscience, Aarhus University, DK-8600 Silkeborg, Denmark; ej@bios.au.dk
7 Sino-Danish Centre for Education and Research (SDC), Beijing 100070, China
8 Limnology Laboratory, Department of Biological Sciences and Centre for Ecosystem Research and Implementation, Middle East Technical University, Ankara 06800, Turkey
9 State Key Laboratory of Lake Science and Environment, Institute of Geography and Limnology, Chinese Academy of Sciences, Nanjing 210008, China
* Correspondence: zliu@niglas.ac.cn (Z.L.); wetlandxfz@163.com (X.Z.)
† These authors contributed equally to this work.

**Abstract:** The presence of omnivorous fish is known to affect aquatic ecosystems, including water quality. The effect, however, depends on the species in question, and our knowledge is limited on the effect of omnivorous crucian carp (*Carassius auratus*), a common and often the most numerous fish species in eutrophic shallow lakes in China. We conducted a 70-day outdoor experiment in mesocosms with and without crucian carp to examine whether this species adversely affects water quality by increasing the levels of total nitrogen (TN) and total phosphorus (TP), thereby stimulating the biomass of phytoplankton and increasing water turbidity. Compared with carp-free controls, the presence of crucian carp resulted in higher TN and TP in the water column, greater phytoplankton biomass and lower periphyton biomass, measured as chlorophyll *a*. Total suspended solids (TSS) also increased in the presence of fish. We conclude that crucian carp can increase TN and TP, enhance phytoplankton biomass, and increase water turbidity, thereby contributing significantly to the deterioration of the water quality. In addition to controlling external nutrient loading, the removal of crucian carp may help to improve water quality in warm shallow eutrophic lakes.

**Keywords:** crucian carp; nutrient; phytoplankton; periphyton; sediment; water quality

## 1. Introduction

Fish are major consumers in many aquatic ecosystems and can have large effects on aquatic ecosystems, including effects on community structure and trophic state [1,2]. Many species of omnivorous fish are easy to introduce and potentially invasive [3,4], and such introductions have led to significant increases in these species in many lakes around the world, often to the point where they have become dominant. The ability

of omnivorous fish to feed opportunistically on organic particles, pelagic and benthic algae, small fish and benthic animals, weeds and seeds [5] can cause a variety of ecological problems in freshwater ecosystems, including serious detrimental impacts on water quality [6,7].

A significant body of research has demonstrated a variety of ways in which omnivorous fish influence shallow lake systems by modifying nutrient levels and turbidity, boosting the growth of phytoplankton and suppressing zooplankton. The disturbance of sediments associated with benthic feeding of omnivorous fish may increase the turbidity of water [8,9] and facilitate the release of nutrients in shallow lakes [10–12]. Common carp (*Cyprinus carpio*) may cause the resuspension of sediment and nutrient release through feeding on and expelling sediment [13]. Excretion by omnivorous fish can increase nutrient concentrations and stimulate phytoplankton growth [3,9], which also leads to reduced light intensity at the sediment surface [14–16]. Matsuzaki et al. [17] and Fischer et al. [18] confirmed that excretion by common carp was an important nutrient source for phytoplankton. In addition, omnivorous fish can influence the community structure of zooplankton [19,20] that play a key role in the environmental state of lakes by grazing on phytoplankton [21].

Despite the negative effects described above, studies by Qiu et al. [22] and Roberts et al. [23] revealed that nutrient and chlorophyll *a* (chl *a*) concentrations in the water column were not affected by omnivorous fish, and Fletcher et al. [24] found that omnivorous fish did not increase turbidity. These various patterns show that the implications of omnivorous fish for water quality are not fully understood and warrant further studies.

The crucian carp (*Carassius auratus*) is a freshwater omnivorous species that feeds on organic detritus, periphyton, phytoplankton, zooplankton, small benthic animals and other organic matter. It is native to China, where it is common in many lakes and rivers [5]. The species has expanded its range via deliberate and accidental introduction to freshwater lakes, rivers and reservoirs in many countries, but, despite being widespread, the effects of the species on water quality are still unclear [25–27]. We hypothesized that the presence of crucian carp would have a negative impact on water quality in shallow lakes by increasing nutrient levels, enhancing phytoplankton growth and decreasing water clarity. To test this hypothesis, an outdoor mesocosm experiment was conducted in which water quality parameters were compared between mesocosms containing crucian carp and carp-free controls. The results of our study may be of use to managers concerned with the effects of omnivorous fish on the water quality of aquatic ecosystems [28].

## 2. Materials and Methods

### 2.1. Experimental Material

Sediment (total nitrogen (TN) = 1.13 mg g$^{-1}$; total phosphorus (TP) = 0.56 mg g$^{-1}$) was collected from a shallow eutrophic lake in Guangzhou, China [27], and the water used was rainwater. Prior to the test, the sediment was sieved through a 0.5-cm sieve to remove coarse debris and mixed to ensure uniformity. The crucian carps were purchased from a local market in Guangzhou City and were kept in 200-L tanks for two weeks before the experiment. The fish were not fed during this period, or during the experiment.

### 2.2. Experimental Design

Eight circular mesocosms (upper diameter = 57 cm, bottom diameter = 46 cm, height = 82 cm) were established at Jinan University. Processed sediments were added to each mesocosm in a ~10-cm-thick layer, after which the mesocosms were filled with rainwater (TN = 0.993 mg·L$^{-1}$, TP = 0.016 mg·L$^{-1}$) and exposed to natural sunlight. Once prepared, the mesocosms were equilibrated for about two weeks. After that time, the nutrient concentrations were 1.14 mg·L$^{-1}$ for TN and 0.07 mg·L$^{-1}$ for TP and the increased nutrients were probably released from the sediment, especially soluble reactive phosphorus (SRP). An artificial substrate made of plastic imitating a piece of leaf (22.0 cm length, 1.2 cm width) was placed on the surface of the sediment in each mesocosm to allow for periphyton growth. One crucian

carp (average length $15.3 \pm 0.4$ cm and weight $51.5 \pm 2.4$ g) was added to each of the four mesocosms as crucian carp treatment, while the other four mesocosms remained without fish as controls. To prevent the fish from jumping out, all the mesocosms were covered with a thin wire mesh (10 mm mesh size). Nitrogen and phosphorus were added weekly to each mesocosm as 1.5 mg $N{\cdot}L^{-1}{\cdot}week^{-1}$ in the form of $KNO_3$ and 0.1 mg $P{\cdot}L^{-1}{\cdot}week^{-1}$ as $NaH_2PO_4$, respectively [29]. During the experiment, water levels were maintained by the addition of rainwater, if necessary. This was not a major contributor to nutrient additions in the mesocosms. The experiment was conducted from 15 June to 26 August 2018, during which time the mesocosms were exposed to natural sunlight. Our study was limited to the summer season as we expected fish activity to be higher in summer than in other seasons [30] and fish effects, therefore, were more detectable. However, additional studies in other seasons are needed to confirm whether this expectation is correct.

### 2.3. Sampling

Light intensity was measured by an underwater irradiance illuminometer at a water depth of 50 cm at approximately 12 noon, every two weeks. Water samples (1 L) were taken from 10 to 20 cm under the water surface in the middle of each mesocosm with a clean polyethylene bottle every two weeks for analysis of total nitrogen (TN) and total phosphorus (TP), total suspended solids (TSS) and phytoplankton biomass (chl *a*). Chl *a* was determined from material retained on glass microfiber filters (Whatman GF/C) and extracted in 90% acetone/water solution within 24 h, after which the concentration was measured in a spectrophotometer [31]. TN and TP were determined according to American Public Health Association (APHA) [32]. TSS was recorded as the residue retained from 500 mL water passing a GF/C grade filter dried at 108 °C for 2 h.

Since it was difficult to quantitatively collect periphyton grown on the wall of the mesocosms and on sediment, we collected periphyton growing on the artificial leaf in each mesocosm by brushing with a soft brush. Chl *a* content was used as a proxy for biomass and measured by a spectrophotometer. After the collection of water and periphyton samples, nutrients were added to each mesocosm and a new artificial substrate was placed in each mesocosm.

On day 0 (initial treatment) and day 70 (the end of the experiment), the cladocerans in each mesocosm were collected by filtering 10-L water samples through a 64-μm mesh, after which the sample was fixed and stored in 8% formaldehyde. Cladocerans were identified to species according to Chiang and Du [33], and their density was calculated. We gave special attention to large cladocerans as they are recognized as highly effective grazers of phytoplankton and are readily consumed by omnivorous fish. At the end of the experiment, the weight of the fish was also recorded.

### 2.4. Statistical Analyses

The effects of crucian carp on nutrient concentrations, the biomass (chl *a*) of phytoplankton and periphyton, TSS and the light intensity at the sediment surface were determined using repeated measures analyses of variance (RM-ANOVAs), with time as the repeated factor after checking for normality and homogeneity of variance in the samples. If the assumption of sphericity of the variance–covariance matrices of the RM-ANOVA analyses was violated, the degrees of freedom (df) were corrected (Huynh–Feldt correction), resulting in an adjustment of the significance of the F ratio. One-way ANOVA was performed to detect differences among treatments on each sampling occasion, including the cladoceran density between the start and the end of the experiment. Prior to analysis, if needed, data were log10 transformed to meet the assumptions of normality or homogeneity of variance. All statistical analyses were performed using SPSS 19.0. Data were presented as mean ± SD.

## 3. Results

### 3.1. TN and TP

TN and TP (Figure 1) were generally higher in the crucian carp treatments than in the controls (RM-ANOVAs, treatment effect, $p < 0.05$). TP varied significantly over time (RM-ANOVAs, time effect, $p < 0.05$). TN and TP were higher in the crucian carp treatments than in the controls on every sampling occasion except day 56 and day 70 (one-way ANOVA, treatment effect, $p < 0.05$; Figure 1).

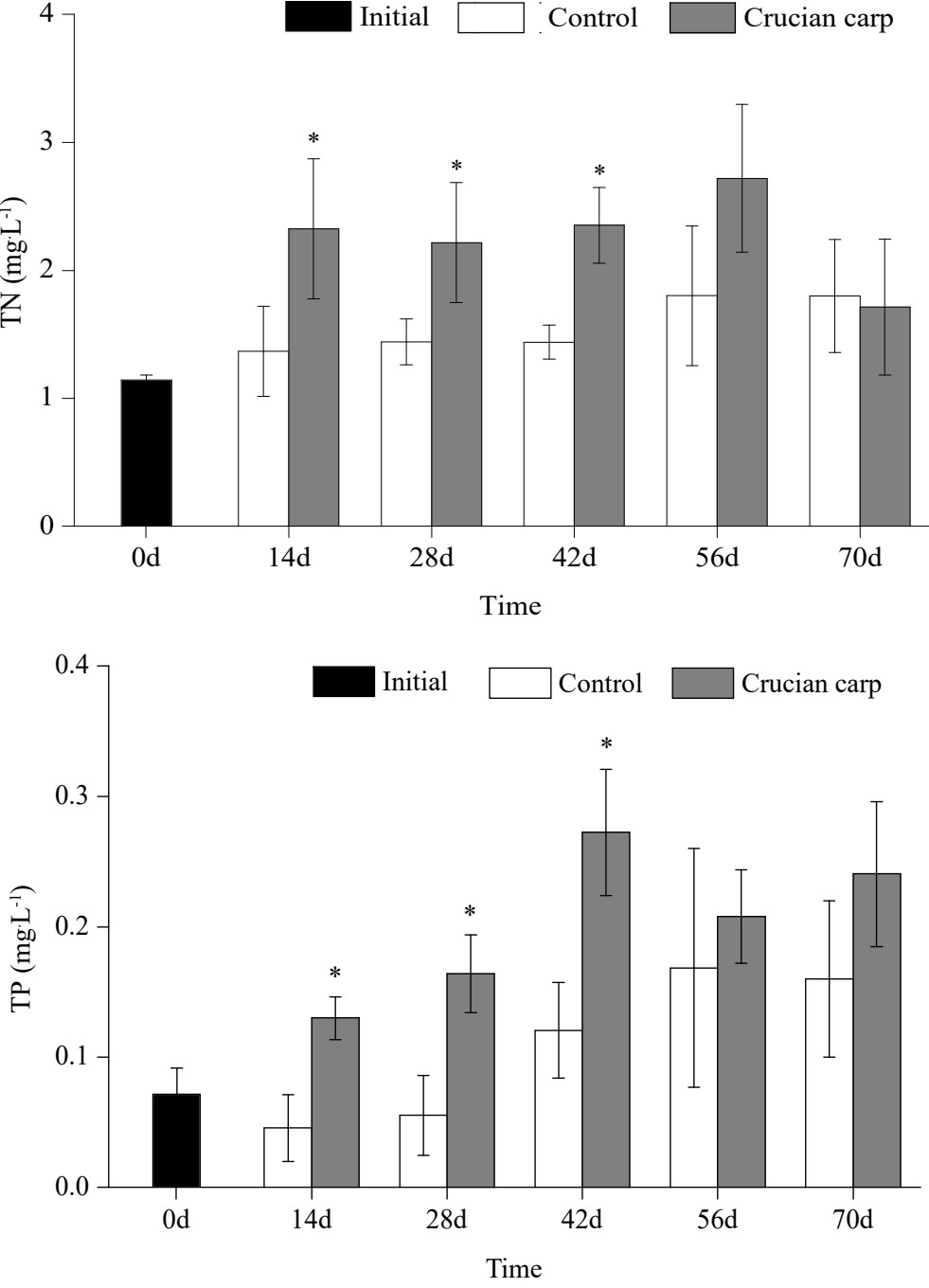

**Figure 1.** Total nitrogen (TN, mean ± SD) and total phosphorus (TP, mean ± SD) in the different treatments over time. Asterisk indicates significant ($p < 0.05$) differences between the crucian carp treatments and the controls on each sampling occasion.

### 3.2. Phytoplankton and Periphyton Biomass

Phytoplankton biomass (chl *a*) was generally higher in the crucian carp treatments than in the controls (RM-ANOVAs, treatment effect, $p < 0.05$). The chl *a* values did not vary significantly with time (RM-ANOVAs, time effect, $p > 0.05$). However, chl *a* was higher in the crucian carp treatments than in the controls on day 14 and day 42 (one-way ANOVA, treatment effect, $p < 0.05$; Figure 2).

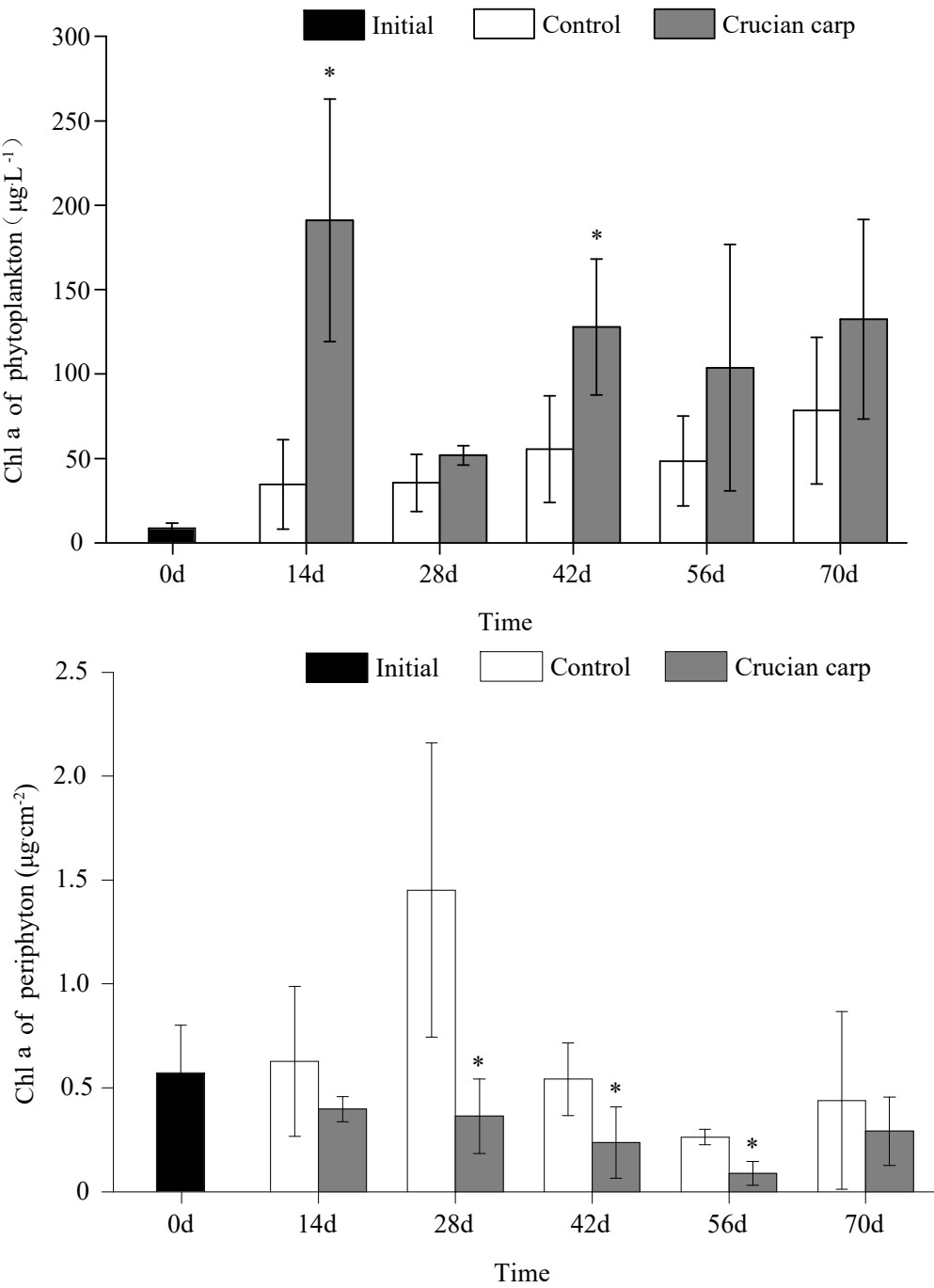

**Figure 2.** Chl *a* of phytoplankton and periphyton (mean ± SD) in the different treatments over time. Asterisk indicates significant ($p < 0.05$) differences between the crucian carp treatments and the controls on each sampling occasion.

In contrast, periphyton biomass (chl *a*) was lower in the crucian carp treatments than in the controls (RM-ANOVAs, treatment effect, $p < 0.05$). The chl *a* values also varied significantly with time

(RM-ANOVAs, time effect, $p < 0.05$), being lower in the crucian carp treatments than in the controls on days 28, 42 and 56 (one-way ANOVA, treatment effect, $p < 0.05$; Figure 2).

### 3.3. TSS and Light Intensity

TSS was higher in the crucian carp treatments than in the controls (RM-ANOVAs, treatment effect, $p < 0.05$) with significant differences observed on every sampling occasion except for day 56 and day 70 (one-way ANOVA, treatment effect, $p < 0.05$, Figure 3).

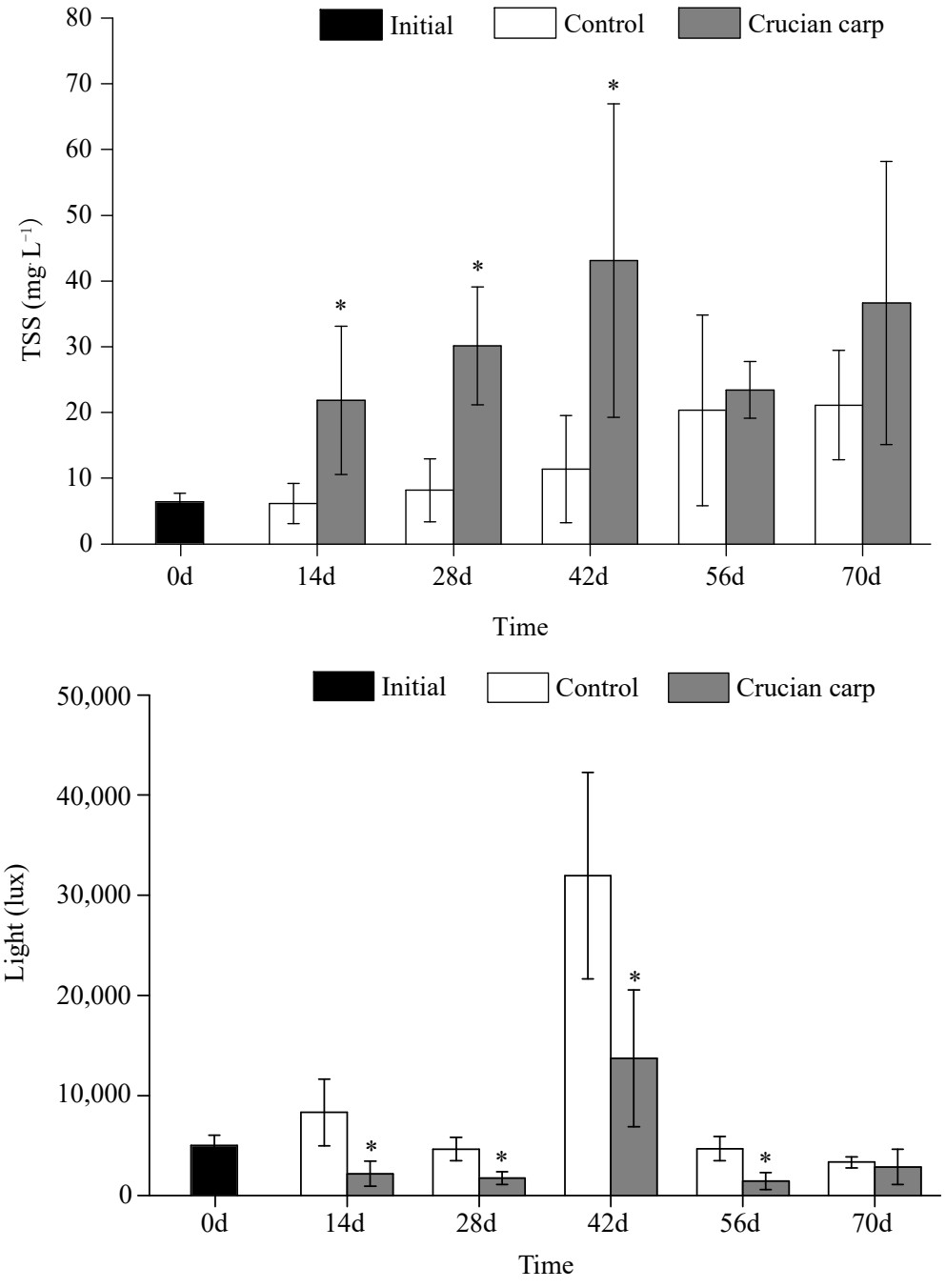

**Figure 3.** Total suspended solids (TSS, mean ± SD) and light intensity (mean ± SD) in the different treatments over time. Asterisk indicates significant ($p < 0.05$) differences between the crucian carp treatments and the controls on each sampling occasion.

Light intensity at the sediment surface was lower in the crucian carp treatments than in the controls (RM-ANOVAs, treatment effect, $p < 0.05$) and declined significantly with time (RM-ANOVAs, time effect, $p < 0.05$). The difference between treatments and controls was apparent on every sampling occasion except day 70 (one-way ANOVA, treatment effect, $p < 0.05$, Figure 3). Note that light intensity should not be compared directly with TSS levels as ambient light levels differed between sampling occasions due to variable cloud cover and haze.

### 3.4. Zooplankton

At the end of the experiment, the density of cladocerans did not differ between fish treatments ($32 \pm 13$ ind·L$^{-1}$) and the controls ($22 \pm 6$ ind·L$^{-1}$) ($p > 0.05$, Figure 4). Unfortunately, the starting density differed significantly in the two treatments being $95 \pm 35$ ind·L$^{-1}$ in fish treatments and $28 \pm 10$ ind·L$^{-1}$ in the controls.

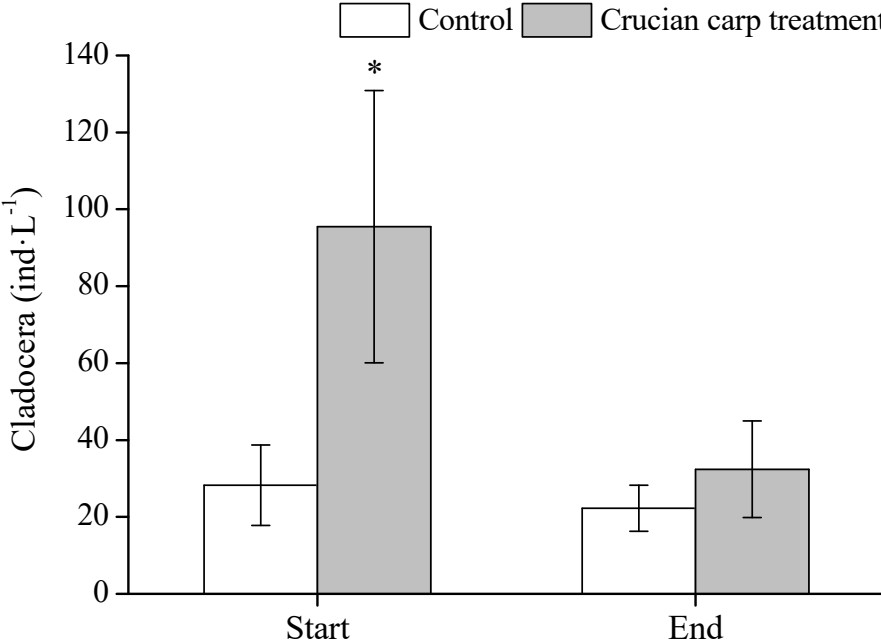

**Figure 4.** Density of cladocerans at the start and at the end of the experiment (mean ± SD) in the different treatments. The asterisk indicates significant ($p < 0.05$) differences.

## 4. Discussion

The presence of crucian carp increased nutrient concentrations, phytoplankton biomass and TSS in our experimental mesocosms, whereas light intensity on the sediment surface and periphyton biomass decreased in the presence of fish, with the overall result that crucian carp were associated with a significant deterioration in water quality.

The behaviour of omnivorous fish and especially those that feed on benthic animals or benthic algae is known to enhance the release of nutrients from sediments [34–38]. The effect occurs partly through bioturbation of the sediment but also by fish excretion of nutrients [39]. The transfer of nutrients from benthic habitats to planktonic habitats has an enhancing effect on phytoplankton growth [35,36,40–42] and may also change the phytoplankton community. Our results are consistent with earlier findings that the presence of omnivorous fish resulted in an overall increase in TN and TP concentrations and an increase in phytoplankton biomass. Zhang et al. [12] observed that omnivorous tilapia (*Oreochromis niloticus*) can also increase TN, TP and phytoplankton biomass (chl *a*) in the water column. Omnivorous crucian carp can thus also accelerate eutrophication.

Sediments may be both "sinks" or "sources" for nutrients, and play important roles in nutrient cycling in shallow aquatic ecosystems and affecting water quality. The increased levels of TN (123.1 ±

78.3 mg/mesocosm on average) and TP (16.4 ± 7.3 mg/mesocosm on average) in the crucian carp treatments likely originated from the sediment released or nutrient additions to the water column by fish bioturbation. Nutrients are also excreted by fish. Note that these excreted nutrients were still derived from the sediment or nutrient additions, as fish did increase in biomass during the course of the experiment from 51.5 ± 2.4 g at the beginning to 56.7 ± 1.9 g at the end of the experiment. Assuming 10% dry/wet of fish biomass, N content of 9.0% and P content of 2.35% of their dry biomass [43], there was 47.3 ± 17.0 mg N and 12.3 ± 4.4 mg P sequestered in the increased fish biomass. Therefore, it is unlikely that they were net contributors of nutrients to the mesocosms. Another possible reason for the increased TN in the water column is nitrogen fixation by cyanobacteria. However, this potential contribution was not evaluated in this experiment. Given that fish increased turbidity, and that N-fixation is a light-dependent process, higher N fixation in the fish mesocosms than in controls is unlikely.

We also found an increase in TSS and a reduction in the light intensity on the sediment surface linked with the presence of crucian carp. Crucian carp can resuspend sediment particles during benthic feeding and thus increase water turbidity and TSS levels in the water [27,44]. Likewise, He et al. [7] found that crucian carp enhanced sediment resuspension and increased TSS in the mesocosms with crucian carp relative to controls. In addition, the fish-induced enhanced growth of phytoplankton increased TSS and reduced light penetration. The loss of light may limit the growth and biomass of periphyton [45]. Furthermore, the grazing pressure exerted by omnivorous fish can be an important factor in reducing periphyton biomass [46–48]. The crucian carp in our study may have contributed to the observed reduction of the periphyton biomass by both increased turbidity and direct grazing. Under eutrophic conditions, light is likely to be a more important factor limiting periphyton growth [49]. However, more research is still needed concerning the effect of fish on periphyton biomass, including both the periphyton growing on the wall of the mesocosms and on the sediment.

Cladocerans are recognized as highly effective grazers of phytoplankton [50,51]. The abundance of cladocerans usually increases with increasing nutrient concentrations due to the enhanced availability of food [52,53]. However, in this study, the number of cladocerans (chiefly *Diaphanosoma paucispinosum*, *Daphnia similoides sinensis* and *Dunhevedia crassa*) in the nutrient-rich fish treatment and the less-rich control were similar at the end of the experiment. Though we mixed the sediment to ensure uniformity in each mesocosm, the number of cladocerans in each mesocosm was higher in the mesocosms receiving crucian carp than in controls after the initial two weeks equilibrium period. Cladocerans have fast turnover rates and could have varied considerably over time during the 70 days of the experiment; thus, the only conclusion we can make here is that cladocerans did not contribute to the difference observed at the end of the experiment. Other studies have, however, found a strong effect of crucian carp on zooplankton [20].

The biomass of omnivorous fish in aquatic ecosystems varies and can be as high as 390–810 g·m$^{-2}$ [54]. The fish biomass used in this study is realistic for shallow lakes of 247 ± 11 g·m$^{-2}$. Omnivorous fish, such as crucian carp, have been widely introduced throughout the world, and our results suggest that the introduction or removal of omnivorous fish will alter ecological processes and functions of an aquatic ecosystem, and its water quality. Elsewhere, the presence or absence of a single species has dramatically altered ecosystem processes [55,56].

## 5. Conclusions

The current study demonstrates that the presence of crucian carp was associated with increased TN, TP, phytoplankton biomass and TSS and with reduced light intensity on the sediment surface, and lower periphyton biomass. These effects are all considered negative for water quality, so our results suggest that the removal of crucian carp may be a useful measure for managers of shallow aquatic ecosystems seeking to improve water quality provided that the nutrient concentrations are sufficiently low to allow for the long-term success of such a restoration.

**Author Contributions:** Conceptualization, Y.H., X.M., Z.L. and X.Z.; investigation, Y.H. and X.Z.; formal analysis: Y.H. and X.Z.; writing—original draft preparation, Y.H. and X.Z.; writing—review and editing, Y.H., X.M., L.G.R.,

W.D.T., J.U., E.J., Z.L. and X.Z.; funding acquisition, X.M. and X.Z. All authors have read and agree to the published version of the manuscript.

**Funding:** This study was funded by National Natural Science Foundation of China (No. 41771100; 41811530056 and 31570456) and Provincial Natural Science Foundation of Guangdong (No. 2016A030313103), China. EJ was supported by the TÜBITAK 2232 outstanding researcher program.

**Acknowledgments:** We thank Anne Mette Poulsen and Amy-Jane Beer for valuable editorial work on this manuscript.

**Conflicts of Interest:** The authors declare no conflict of interest.

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
