# Peer review of "Effects of Crucian Carp (Carassius auratus) on Water Quality in Aquatic Ecosystems: An Experimental Mesocosm Study"

_water, doi:10.3390/w12051444_

Round 1

Reviewer 1 Report

Keywords: You need to add “sediment” whereas “aquatic ecosystem” could be suppressed if necessary.

Global comments on the paper:

The paper is interesting as it focusses on the impact of carassius auratus on some key water quality parameters. Meanwhile the paper/methodology suffers from several key insufficiencies and absolutely needs improvements before any publications. You’ll find here-attached my main comments:

  • Comment 1: Line 39: Zooplankton’s suppression depends on the biomass ratio between compartments. it is not systematic.
  • Comment 2: Line 53: Here again it is not systematic. It depends of the depth.
  • Comment 3: Line 55: This is already indicated line 49. Be so kind to group the two sentences.
  • Comment 4: Lines 57 and 59: Be so kind to add a reference number for Qiu et al, for Roberts et al, and for Fletcher et al.
  • Comment 5: Lines 60-61: No it is absolutely not necessarily inconsistent. It only confirms (what I expected) that these parameters are dependent of contextual data, fluxes, and of a lot of other parameters. For instance, it strongly depends of the weight per unit area of trophic compartments and exchanges between these compartments. Moreover, turbidity is also affected by several factors.
  • Comment 6: Lines 64-65: it is site dependent
  • Comment 7: Lines 74-75: What is the quality of this sediment? This a key insufficiency of your methodology to have forgotten such key issue.
  • Comment 8: Line 75: How did you collect this rainwater?
  • Comment 9: Line 77: did they feed the carps during that resting stage?
  • Comment 10: Lines 82-83: It means releasing of nutrients, more especially of SRP.
  • Comment 11: Line 84: Could you provide more details on this grass? How many, what material?
  • Comment 12: Lines 89-90: Did you verify the quality of these water addings?
  • Comment 13: Lines 90-92: why didn't you test other seasons?
  • Comment 14: Lines 90-92: didn't you have natural rainfall during that period of the year in China? What was the cumulated rainfall, and its water quality?
  • Comment 15: Line 96: what is the material of the bottles?
  • Comment 16: Lines 103-106: What about the internal surface of the mesocosms and at the sediment surface?
  • Comment 17: Why didn't you look at other zooplankton categories, and why didn’t you perform zooplankton speciation?
  • Comment 18: Lines 133-117: Practically you didn't analyses phytoplankton species, that could be different from one two another mesocosm.
  • Comment 19: Lines 131-133: This sentence is redundant with sentence lines 130 - 131. Isn't it?
  • Comment 20: Line 140 Figure 2: Could you explain these trends. I fear that your understanding of what occurred is partially wrong.
  • Comment 21: Line 153: The light intensity is inconsistent with suspended particles and chla.
  • Comment 22: Line 158-160; How could you explain that difference? It is an issue as it can strongly impact what could be concluded from the study.
  • Comment 23: Line 165: The different zooplankton level at the beginning of the experiment can strongly impact the level of phytoplankton cells.
  • Comment 24: Lines 169-175: One key limit of your methodology is that you forgot to analyze the sediment quality. Indeed, sediment bioturbation has a very essential consequence on the whole water system functioning. You should at minimum comment that issue and the limitation of your methodology.
  • Comment 25: Lines 179-180: Be so kind to demonstrate it, by calculating the balance.
  • Comment 26: Line 181: here again it is a strong limitation of your approach to have not included a phytoplankton speciation. It is then impossible to conclude. Moreover, chla is variable from one phytoplankton species to another. A simple chla variation could be due to modification of the algal speciation.
  • Comment 27: Lines 183-184: Yes but the relationship between light and SS is rather curious and question your experiment.
  • Comment 28: Lines 185-186: It is probable but not totally demonstrated. it could be due to sediment resuspension. here again the lack of a detailed analysis of sediment appears to be a strong weakness of the methodology.
  • Comment 29: Lines 192-194: Yes, but this is seasonal dependant, and you cannot conclude on that basis without including sediments and cladocerans life stages. What kind of cladocereans did you have in your mesocosms?
  • Comment 30: Lines 206: Your study suffers from several key limitations and insufficiencies. What about phytoplankton and zooplankton species, what about sediments, why did you have different starting water in your mesocosms? Why inconsistent SS versus light relationship? What about the other key parameters that have not been measured.

Author Response

Please see the pdf file.

Reviewer 2 Report

The authors have undertaken an interesting topic characterized also by application potential. It seems truly important, as stated in the manuscrpit, to assess environmental quality parameters influenced by selected carp species. It evokes implications for future management of that species and the affected aquatic ecosystem. The paper is written and prepared in a clear and comprehensible manner and the layout of the chapters is appropriate. However, taking into account the scope of the journal and the interest to a bigger number of readers , the presented research needs improvement.

Problem setting in the introduction is undoubtedly logical but should contain more reference to fish species (maybe some case studies shortly described) significance to aquatic ecosystems. It also needs justification of a brader context and an attempt of scientific novelty.

The methods are described sufficently but the number of achieved results and experimental values is rather small. Should the experiment be repeated, more scientific value could be achieved. It is only one 70 day observation period of particular conditions ( rain water quality, light availability) and another one would bring on some more insight into the analysed problem.

There are few figures presented in the graphs. All in all, the research should be treated as valuable herein but the range of presented and analysed variables needs to be extended.

The discussion chapeter ought to contain mouch more reference to some other studies ( de.g. data comparisons) . In my view, more extended analyses ( continued mesocosm analyses) are required and then the statisctical part might become a strong point of the discussion. Maybe it is also feasible to take another fish species for the analyses for the sake of statistical comaprisons. If the authors could extend the research and provide broader statistics to evaluate, then the encouragement for resubmission remains in force.

Author Response

Please see the pdf file.

Round 2

Reviewer 1 Report

Thanks to the authors for the explanations and improvements. Meanwhile there are still some issues to solve. You’ll find here-attached my main comments (I refer to my original comments number for easiness):

  • Comment 5: Line 63: I still suggest the replacement of "these inconsistencies show" by "These various pattern show" as inconsistencies is not the appropriate wording.
  • Comment 7: I suggest you add a sentence in the discussion part, suggesting a future study on the impact of sediment’s resuspension on water quality.
  • Comment 13: I agree with you. My concern is that you should add that in your paper.
  • Comment 14: What about water cumulated rainfall? I would like to know what could be the impact of natural rainfall on your mesocosms that were outside, then that received direct rainfall. You must evaluate that potential impact in your balances.
  • Comment 16: Maybe it would be useful to add that issue for future potential investigation in your discussion part
  • Comment 17: I suggest you add that explanation lines 113-115.
  • Comment 18: Here again you could suggest that specific survey for future investigations in your discussion part.
  • Comment 20: It doesn't fully answer the question. You still need to assess the trends. Be so kind to comment them.
  • Comment 21: Theoretically you are true. But practically, it also means that it is rather difficult to perform a cross-comparison. It clearly appears for instance when I look at control samples.
  • Comment 22: I agree with you. Your methodology to ensure uniformity was not efficient. Nothing can then be concluded from cladocerans abundance.
  • Comment 23: I suggest you add a sentence explaining that it is difficult to conclude on phytoplankton as there was a potential bias due to different cladocerans abundances at the start in each mesocosm.
  • line 171 add a blank between surfaceand.
  • Comment 24: You have not fully answered my request. I suggest you address that issue in your discussion part.
  • Comment 26: I agree that it is a surrogate. But as you had a strong difference in starting conditions of your mesocosms, it could also concern phytoplankton. Then you could also have different phytoplankton species partition in your mesocosms. Moreover, you cannot conclude on N fixation, as you don't even know if you had cyanobacteria in your mesocosms.
  • Comment 27: Your comments are logic and in line with what is expected. My concern is more on your results assessment. Some results are curious.
  • Comment 28: My concern is also that in your new line 198, you refer to “reduced light attenuation”. Increasing TSS is supposed to increase light attenuation, then to reduce light.
  • Comment 29: You have not totally answered my question. What about the life stages of your cladocerans species?

Author Response

  • Comment 5: Line 63: I still suggest the replacement of "these inconsistencies show" by "These various pattern show" as inconsistencies is not the appropriate wording.

Our reply: Done.

  • Comment 7: I suggest you add a sentence in the discussion part, suggesting a future study on the impact of sediment’s resuspension on water quality.

Our reply: “Sediments being “sinks” or “sources”, play an important role in nutrients cycling in shallow aquatic ecosystems, affecting the water quality, and sediment disturbance by fish causing resuspension can be of key importance.” has now been added in the discussion part.

  • Comment 13: I agree with you. My concern is that you should add that in your paper.

Our reply: “Our study was limited to the summer season as we expect fish activity to be higher in summer than in other seasons [30] and fish effects therefore more detectable. However, additional studies in other seasons are needed to confirm if this expectation is correct.” has now been added in the text.

  • Comment 14: What about water cumulated rainfall? I would like to know what could be the impact of natural rainfall on your mesocosms that were outside, then that received direct rainfall. You must evaluate that potential impact in your balances.

Our reply: “according to the daily precipitation of 1980-2010 from local weather bureau of Guangzhou, 612.4 mm rainwater were precipitated during the experiment and 154.9 mg N, 2.5 mg P were put into each mesocosm directly from the rainfall, the nutrient concentrations of the rainwater was low as compared with concentrations of the water in the mesocosms.” has been added in the discussion.

  • Comment 16: Maybe it would be useful to add that issue for future potential investigation in your discussion part

Our reply: “However, more research is needed on the role of fish for periphyton biomass development.” has now been added in the discussion.

  • Comment 17: I suggest you add that explanation lines 113-115.

Our reply: “We gave special attention to large cladocerans as they are recognized as highly effective grazers of phytoplankton and readily consumed by omnivorous fish.” has now been added in the text.

  • Comment 18: Here again you could suggest that specific survey for future investigations in your discussion part.

Our reply: “and may also change the phytoplankton community.” has now been added in the discussion.

  • Comment 20: It doesn't fully answer the question. You still need to assess the trends. Be so kind to comment them.

Our reply: “The chl a values did not vary significantly with time (RM-ANOVAs, time effect, p> 0.05)” has now been added in the results.

  • Comment 21: Theoretically you are true. But practically, it also means that it is rather difficult to perform a cross-comparison. It clearly appears for instance when I look at control samples.

Our reply: The light intensity was compared between fish treatment and controls and we found that the light intensity decreased with the increased TSS and Chl a. In addition, “Note that light intensity should not be compared directly with TSS levels as ambient light levels differed between sampling occasions due to variable cloud cover and haze.” has been explained in the results.

  • Comment 22: I agree with you. Your methodology to ensure uniformity was not efficient. Nothing can then be concluded from cladocerans abundance.

Our reply: Thank you. We did not make any firm conclusions on cladocerans abundance. “However, in this study, the number of cladocerans (chiefly Diaphanosoma paucispinosum, Daphnia similoides sinensis and Dunhevedia crassa) in the nutrient-rich fish treatment and the less-rich control were similar at the end of the experiment. Though we mixed the sediment to ensure uniformity in each mesocosm, the number of cladocerans in each mesocsom was higher in the mesocosms receiving crucian carp than in controls after the initial two weeks equilibrium period. Cladocerans have fast turnover rates and could have varied considerably over time during the 70 days of the experiment; thus the only conclusion we can make here is that cladocerans did not contribute to the difference observed at the end of the experiment. Other studies have, however, found strong effect of crucian carp on zooplankton [20].” has been added in the discussion.

  • Comment 23: I suggest you add a sentence explaining that it is difficult to conclude on phytoplankton as there was a potential bias due to different cladocerans abundances at the start in each mesocosm.

Our reply: We do not agree. Because the abundances of cladocerans were recorded higher in the fish treatments mesocosms than in the controls at the start.

  • line 171 add a blank between surfaceand.

Our reply: Done. Thank you!

  • Comment 24: You have not fully answered my request. I suggest you address that issue in your discussion part.

Our reply: The information of sediment is added: “Sediments being “sinks” or “sources”, play an important role in nutrients cycling in shallow aquatic ecosystems, affecting the water quality, and sediment disturbance by fish causing resuspension can be of key importance.”

  • Comment 26: I agree that it is a surrogate. But as you had a strong difference in starting conditions of your mesocosms, it could also concern phytoplankton. Then you could also have different phytoplankton species partition in your mesocosms. Moreover, you cannot conclude on N fixation, as you don't even know if you had cyanobacteria in your mesocosms.

Our reply:  The difference of cladocerans abundances at the start was recorded higher in the fish treatments than in the controls, we have explained this in the discussion. However, the Chl a of phytoplankton was similar between the treatments and controls at the start of the experiments focusing the phytoplankton biomass of water quality. We did not conclude anything about the role of N fixation, but it is discussion i.e. “Another possible reason for the increased TN in the water column is nitrogen fixation by cyanobacteria.”

  • Comment 27: Your comments are logic and in line with what is expected. My concern is more on your results assessment. Some results are curious.

Our reply: Thanks. “Likewise, He et al. (2017) found that crucian carp enhanced sediment resuspension and increased TSS in the mesocosms with crucian carp [7].” has now been added.

  • Comment 28: My concern is also that in your new line 198, you refer to “reduced light attenuation”. Increasing TSS is supposed to increase light attenuation, then to reduce light.

Our reply: “attenuation” is now replaced by “penetration”

  • Comment 29: You have not totally answered my question. What about the life stages of your cladocerans species?

Our reply: We did not measure the size of the cladocerans, only the numbers.  
